# Reduced marine phytoplankton sulphur emissions in the Southern Ocean during the past seven glacials

K. Goto-Azuma [1,2], M. Hirabayashi[1], H. Motoyama [1,2], T. Miyake[1], T. Kuramoto [1,8], R. Uemura [1,9], M. Igarashi[1], Y. Iizuka[3], T. Sakurai [1,10], S. Horikawa[3,11], K. Suzuki [4], T. Suzuki [5], K. Fujita [6], Y. Kondo [1], S. Hattori [7] & Y. Fujii[1]

Marine biogenic sulphur affects Earth's radiation budget and may be an indicator of primary productivity in the Southern Ocean, which is closely related to atmospheric $CO_2$ variability through the biological pump. Previous ice-core studies in Antarctica show little climate dependence of marine biogenic sulphur emissions and hence primary productivity, contradictory to marine sediment records. Here we present new 720,000-year ice core records from Dome Fuji in East Antarctica and show that a large portion of non-sea-salt sulphate, which was traditionally used as a proxy for marine biogenic sulphate, likely originates from terrestrial dust during glacials. By correcting for this, we make a revised calculation of biogenic sulphate and find that its flux is reduced in glacial periods. Our results suggest reduced dimethylsulphide emissions in the Antarctic Zone of the Southern Ocean during glacials and provide new evidence for the coupling between climate and the Southern Ocean sulphur cycle.

[1] National Institute of Polar Research, Research Organization of Information and Systems, 10-3 Midori-cho, Tachikawa, Tokyo 190-8518, Japan. [2] Department of Polar Science, Graduate University for Advanced Studies (SOKENDAI), 10-3 Midori-cho, Tachikawa, Tokyo 190-8518, Japan. [3] Institute of Low Temperature Science, Hokkaido University, Kita-19, Nishi-8, Kita-ku, Sapporo 060-0819, Japan. [4] Faculty of Science, Shinshu University, 3-1-1 Asahi, Matsumoto 390-8621, Japan. [5] Faculty of Science, Yamagata University, 1-4-12 Kojirakawa-cho, Yamagata 990-8560, Japan. [6] Graduate School of Environmental Studies, Nagoya University, Furo-cho, Chikusa-ku, Nagoya 464-8601, Japan. [7] Department of Chemical Science and Engineering, School of Materials and Chemical Technology, Tokyo Institute of Technology, Yokohama 226-8502, Japan. [8] Present address: Department of Human Development, School of Humanities and Culture, Tokai University, 4-1-1 Kitakaname, Hiratsuka 259-1292, Japan. [9] Present address: Graduate School of Environmental Studies, Nagoya University, Furo-cho, Chikusa-ku, Nagoya 464-8601, Japan. [10] Present address: Civil Engineering Research Institute for Cold Region, Public Works Research Institute, 1-3-1-34, Hiragishi, Toyohira-ku, Sapporo 062-8602, Japan. [11] Present address: Earthquake and Volcano Research Center, Graduate School of Environmental Studies, Nagoya University, Furo-cho, Chikusa-ku, Nagoya 464-8601, Japan. Correspondence and requests for materials should be addressed to K.G.-A. (email: kumiko@nipr.ac.jp)

Dimethylsulphide (DMS) emitted from oceanic phytoplankton plays an important role in controlling concentrations of sulphate ($SO_4^{2-}$) aerosols, which can act as cloud condensation nuclei (CCN)[1–3]. Changes in CCN would influence cloud albedo, a key parameter of radiative forcing[1–3]. Increased $SO_4^{2-}$ can thus cool the Earth by indirect forcing, in addition to direct forcing owing to increased scattering of solar radiation[3]. To understand these effects, DMS emissions and their links to climate should be evaluated in a pristine environment[2]. DMS and its oxidation products, $SO_4^{2-}$ and methanesulphonate ($CH_3SO_3^-$, hereafter MSA), are also indicators of primary productivity in the Southern Ocean (SO), which is important because they are closely related to atmospheric $CO_2$ variability through the biological pump[4]. $SO_4^{2-}$ and MSA in Antarctic ice cores are therefore useful tools for investigating links between the sulphur cycle and climate.

High concentrations of non-sea-salt (nss) $SO_4^{2-}$ measured in glacial samples from Vostok ice core drilled in East Antarctica[5] (Supplementary Fig. 1) have been interpreted as evidence of enhanced oceanic DMS emissions during glacials, assuming that $nssSO_4^{2-}$ is mainly of marine biogenic DMS origin. A subsequent study on the same ice core[6] reports increased MSA concentrations in addition to $nssSO_4^{2-}$, further supporting the interpretation of [5] because MSA originates solely from DMS, whereas $nssSO_4^{2-}$ can come from other sources[7,8]. Based on these results, DMS emissions and hence $nssSO_4^{2-}$ have been believed to exert positive feedback on climate. However, more recent studies refute the positive feedback hypothesis[7,8], showing that MSA is modified post-depositionally in the Antarctic interior where accumulation rates are low and does not represent DMS production around Antarctica. Furthermore, two deep ice cores drilled at Dome C (EDC) and Dronning Maud Land (EDML) in East Antarctica (Supplementary Fig. 1) show little change in $nssSO_4^{2-}$ flux over glacial/interglacial cycles[7–9], while concentrations increase during glacials. Wolff et al.[7] point out that increased $nssSO_4^{2-}$ concentrations in ice cores from sites with low accumulation rates (e.g., Vostok, EDC, EDML) are mainly caused by decreased accumulation rates in glacials, and can therefore not be interpreted as evidence of increased atmospheric $nssSO_4^{2-}$. The nearly constant $nssSO_4^{2-}$ fluxes at EDC and EDML, which face the Indian and Atlantic Ocean sectors of the SO, respectively, have been interpreted to reflect stable DMS emissions and hence stable marine biogenic productivity in the Antarctic Zone (AZ) of the SO over glacial cycles, assuming that the major source of $nssSO_4^{2-}$ is DMS[7–9]. In contrast, marine sediment records show that export production decreases in the AZ during glacials but increases further north in the Sub-Antarctic Zone (SAZ) of the SO[4]. This implies reduced primary productivity in the AZ but increased primary productivity in the SAZ during glacials. The disparity between ice and marine core records has been attributed to differences in marine organisms that contribute to these records[7].

The stable sulphur isotopic composition of $SO_4^{2-}$ ($\delta^{34}S$) provides a useful signature of its origins[10–12]. The $\delta^{34}S$ data measured from EDC and Vostok ice cores suggest 4–6‰ lower $\delta^{34}S$ for the last glacial than for the Holocene and last interglacials, although the data are scattered and sparse[11]. This has been attributed to isotopic fractionation during transport, as terrestrial contribution of $SO_4^{2-}$ has been assumed to be small[11]. However, surface snow samples from a latitudinal transect between a coastal station (Syowa) and an interior site (Dome Fuji, hereafter DF, Supplementary Fig. 1) show remarkably uniform $\delta^{34}S$ in East Antarctica[13]. The results suggest that net isotopic fractionation during long-range transport is insignificant in East Antarctica and thus $\delta^{34}S$ in the ice cores from the East Antarctic interior can be used to infer source contributions. Lower $\delta^{34}S$ values in the

last glacial[11] might be due to an increased contribution of terrestrial $SO_4^{2-}$ originating from increased terrestrial dust[7,8]. Consequently, little change in the $nssSO_4^{2-}$ flux over glacial/interglacial cycles[7–9] can be caused by increased terrestrial sulphate and decreased marine biogenic sulphate.

In this study, we propose this alternative interpretation of $nssSO_4^{2-}$ flux and make a revised calculation of DMS-derived sulphate, using new ice core records obtained at DF, spanning the last 720,000 years[14,15]. On the basis of the revised calculation, we compare the DMS-derived sulphate record from DF with those from EDC and EDML. We find that DMS-derived sulphate fluxes decrease in glacials, which indicates reduced DMS emissions in the AZ of the SO. This suggests that primary production, as well as export production, decreases during glacials, which is consistent with marine sediment records[4].

## Results

**Flux variability and potential sources of $nssSO_4^{2-}$.** We calculated $nssCa^{2+}$ and $nssSO_4^{2-}$ from $Ca^{2+}$, $Na^+$, and $SO_4^{2-}$ concentrations[7–9,16] (Supplementary Figs. 2a–c). Low accumulation rates ($<30$ kg m$^{-2}$ yr$^{-1}$ in the present day and $<50$ kg m$^{-2}$ yr$^{-1}$ throughout the last 720,000 years) (Supplementary Fig. 2d) at DF[14] indicate that the dominant process for aerosol deposition is dry deposition and that the flux, rather than the concentration in ice, better represents the changes in atmospheric aerosol concentration[7,16]. The flux of $nssCa^{2+}$ at DF covaries with that at EDC[7,8] and EDML[16], indicating high and low values during glacials and interglacials, respectively (Fig. 1, Supplementary Fig. 3); fluxes at DF are 2.0 and 0.6 times those at EDC and EDML, respectively.

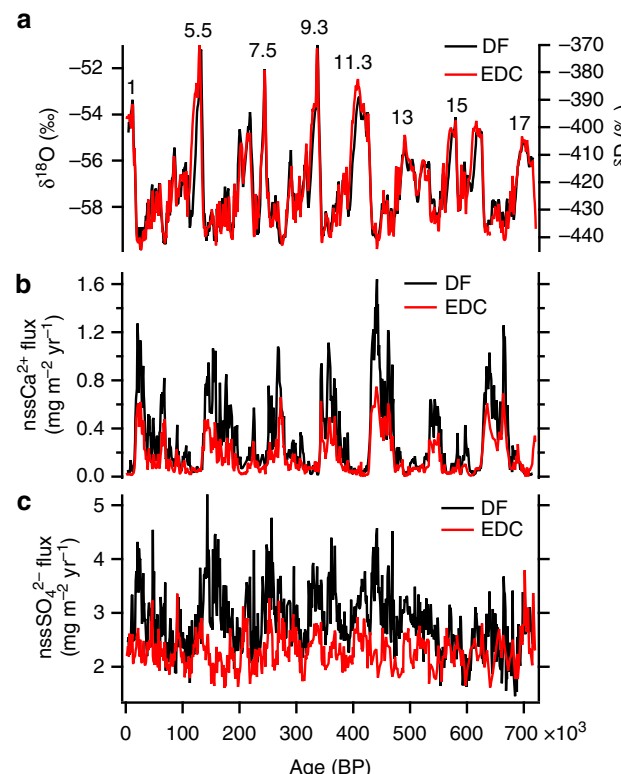

**Fig. 1** Temperature proxies and ion fluxes at Dome Fuji (DF) and Dome C (EDC). **a** The $\delta^{18}O$ (DF)[14] and $\delta D$ (EDC)[7,8] records averaged over 1000 years. Marine isotope stage numbers for interglacials are also shown. **b** Fluxes of $nssCa^{2+}$ at DF and EDC averaged over 1000 years. **c** Fluxes of $nssSO_4^{2-}$ at DF and EDC averaged over 1000 years. See Methods for DF chronology and flux calculations. The EDC fluxes are plotted on the AICC12 timescale[53,54] using previously published ion data[7–9,16] and accumulation rates[53,54]

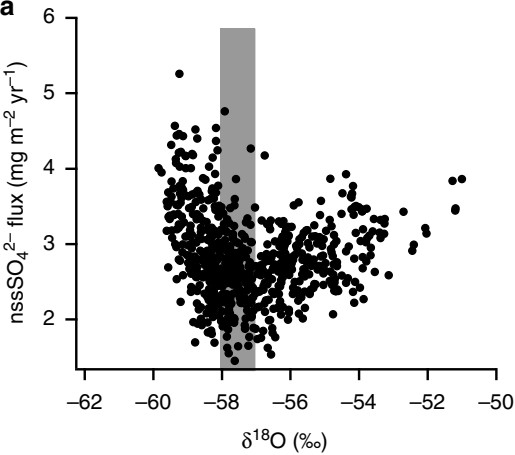

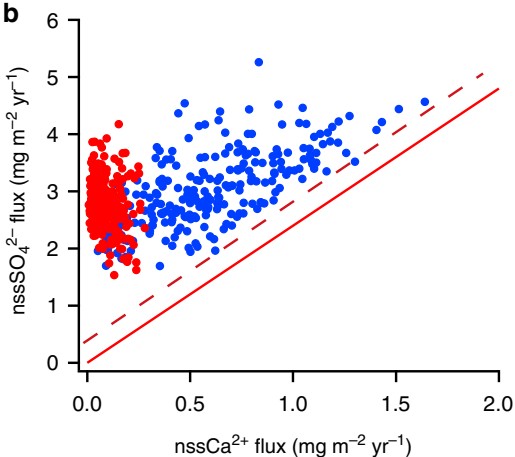

**Fig. 2** Variability of nssSO$_4^{2-}$ flux at Dome Fuji (DF). **a** DF nssSO$_4^{2-}$ flux plotted against DF δ$^{18}$O[14,15]. Data points represent 1000-year averages. Before averaging, the δ$^{18}$O depths that differ from the ion data depths have been interpolated to match. Gray bar indicates the lower threshold of δ$^{18}$O (−58‰), below which the nssSO$_4^{2-}$ flux decreases with δ$^{18}$O, and the upper threshold (−57‰), above which the nssSO$_4^{2-}$ flux increases with δ$^{18}$O. **b** DF nssSO$_4^{2-}$ flux plotted against DF nssCa$^{2+}$ flux. Data points represent 1000-year averages. The slope of the solid red line (m = 2.4) represents the stoichiometric mass ratio of Ca/SO$_4$ as CaSO$_4$. The dashed red line shows the lower bound of the nssSO$_4^{2-}$ flux data with m = 2.4. Red and blue dots represent the data for warm and cold periods, respectively, corresponding to the δ$^{18}$O values above and below the thresholds

The dominant source of nssCa$^{2+}$ is terrestrial dust[7–9,16] and South America is a major source region for dust deposited in the Antarctic interior[17–19]. Different nssCa$^{2+}$ fluxes in three records can be explained by their different distances from the South American source region[16]. Contrary to previous studies on EDC and EDML cores, the nssSO$_4^{2-}$ flux at DF is not constant (Figs. 1 and 2). The flux increases as δ$^{18}$O (a proxy for temperature at DF[14,15]) decreases below approximately −58‰, and increases when δ$^{18}$O is above approximately −57‰.

Potential sources of nssSO$_4^{2-}$ are marine biogenic DMS, volcanic sulphate, and terrestrial dust[7,8]. With the exception of a few years following large volcanic eruptions, the volcanic input is estimated to be less than 10% of the present-day and Holocene sulphate budgets[7]. Oceanic DMS was previously regarded as the dominant nssSO$_4^{2-}$ source over glacial/interglacial cycles, with only a small input from terrestrial sources[7–9]. However, terrestrial sulphate can be a major source in glacials when the amount of dust increases. Fluxes of nssCa$^{2+}$ and nssSO$_4^{2-}$ at DF are

correlated during cold periods when δ$^{18}$O is below approximately −58‰. The scatter plot of nssSO$_4^{2-}$ against nssCa$^{2+}$ (Fig. 2b) shows a lower bound whose slope is close to the stoichiometric ratio for CaSO$_4$, indicating that a large proportion of nssSO$_4^{2-}$ during cold periods exists as CaSO$_4$. This same feature is reported for EDML and a similar but weaker correlation between nssCa$^{2+}$ and nssSO$_4^{2-}$ fluxes is reported for EDC[20]. These observations suggest that a large proportion of nssSO$_4^{2-}$ exists as CaSO$_4$ during cold periods at EDC and EDML[20], as well as DF. Micro-Raman spectroscopic analysis of DF samples from the Last Glacial Maximum (LGM) suggests that a large proportion of Ca$^{2+}$ exists as gypsum (CaSO$_4$·2H$_2$O)[21]. Furthermore, analyses of DF samples from the LGM using scanning electron microscopy/energy-dispersive X-ray spectroscopy (SEM-EDS) show that the majority of Ca$^{2+}$ originates from CaSO$_4$[22]. Although the SEM-EDS analyses indicate that a large fraction of the particles consists of silicate minerals containing Ca, they do not dissolve in water. The majority of Ca$^{2+}$ measured in this study using ion chromatography (see Methods) should therefore originate from CaSO$_4$.

CaSO$_4$ in DF core could originate from two potential sources. First is primary gypsum, i.e., terrestrial gypsum transported from arid source regions as dust[8,9,20,23]. Second is secondary gypsum formed by the reaction of CaCO$_3$, one of the major components of terrestrial dust, with marine biogenic H$_2$SO$_4$ or SO$_2$[20,24,25] during dust transport. If primary gypsum is dominant, the major source of nssSO$_4^{2-}$ during cold periods should be dust, not marine biogenic sulphate. But if secondary gypsum is dominant, the major source of nssSO$_4$ should be marine biogenic sulphate. So far, secondary gypsum has been considered dominant, assuming limited fractions of terrestrial sulphate[7,9,11]. A mean sediment SO$_4^{2-}$/Ca$^{2+}$ ratio of 0.1[9] or 0.18 observed for soils[10,12] is often referred to as a basis of a small terrestrial contribution. To define an uppermost limit,[9] uses a ratio of 0.5 observed in Sharan dust plumes and suggests a maximum terrestrial contribution of only 16%. However, ratios are highly variable and source-dependent[8].

Although the compositions of South American minerals are poorly documented, there is some evidence supporting the hypothesis that SO$_4^{2-}$/Ca$^{2+}$ ratios could be much higher than previously assumed. Soil samples from northeastern and central Patagonia, one of the potential source regions, show SO$_4^{2-}$/Ca$^{2+}$ ratios close to or larger than the stoichiometric ratio of CaSO$_4$[26,27]. Because some of these soil samples have high Na$^+$/Ca$^{2+}$ and Mg$^{2+}$/Ca$^{2+}$ ratios (often much higher than 1), which is not the case for nssNa$^+$/nssCa$^{2+}$ and nssMg$^{2+}$/nssCa$^{2+}$ in the Antarctic ice samples (see [28] for nssNa$^+$/nssCa$^{2+}$ and Supplementary Discussion for nssMg$^{2+}$/nssCa$^{2+}$), such soils may not be the source for CaSO$_4$ in the Antarctic ice cores. However, high SO$_4^{2-}$/Ca$^{2+}$ ratios reported in Patagonia cast doubt on the assumption that the SO$_4^{2-}$/Ca$^{2+}$ ratio of 0.5 observed in Sharan dust plumes gives an uppermost limit[9].

Gypsum is a major mineral in evaporites[29]. A distribution of evaporites has been reported in wide regions of South America[30]. Large areas of the Puna-Altiplano, one of the potential source regions of dust deposited in the Antarctic interior[19], are covered by salt-lake beds[19], which could be sources of gypsum-rich evaporites, although the compositions of these lakes are poorly documented and could vary substantially[29]. Giant evaporite belts dominated by halite and gypsum are also found in this region[31]. Puna evaporites are uniquely characterized by scarce carbonates, whereas sulphates and chlorides are abundant[31]. Furthermore, ion ratios nssCl$^-$/nssNa$^+$ and nssNa$^+$/nssCa$^{2+}$ estimated from EDC core suggest a significant contribution of halides mobilized from continental evaporite deposits[28]. The reaction of CaCO$_3$ with H$_2$SO$_4$ and SO$_2$ is slow[20,25,32,33], and only a partial neutralization of clay or carbonate particles has been observed

even in areas where $SO_2$ and $H_2SO_4$ concentrations are greatly enhanced by volcanic contributions[20,34]. If this can also be extended to the different conditions over the Southern Ocean, then terrestrial gypsum would be needed to explain the relationship between $nssCa^{2+}$ and $nssSO_4^{2-}$ in Antarctic ice cores and may be a major $CaSO_4$ source. To validate our idea, it will be important to establish what source areas could provide such a gypsum-rich source of dust.

**Revised calculations of DMS-derived sulphate.** To calculate the flux of DMS-derived $nssSO_4^{2-}$, the contribution of terrestrial sulphate should be removed. We first subtract the terrestrial $nssSO_4^{2-}$ fraction as a case for a maximum contribution of terrestrial gypsum to $nssSO_4^{2-}$ flux. Assuming that the majority of $nssCa^{2+}$ originates from terrestrial sulphate and that $nssCa^{2+}$ is a major terrestrial cation, we make a first-order estimate of the marine biogenic sulphate flux by subtracting the $CaSO_4$ contribution ($nssCa^{2+}$ multiplied by 2.4, the stoichiometric mass ratio of $SO_4/Ca$ for $CaSO_4$) from the total $nssSO_4^{2-}$ flux. The residual $nssSO_4^{2-}$ is thus dominated by sulphate in the form of $H_2SO_4$ and/or $Na_2SO_4$. Both $H_2SO_4$ and $Na_2SO_4$ originate from DMS; the former is directly produced from DMS, whereas the latter is produced by the reaction between DMS-derived $H_2SO_4$ and NaCl[35] (sea salt and/or terrestrial). The residual $nssSO_4^{2-}$ flux, a revised marine biogenic sulphate flux, co-varies with the temperature proxy $\delta^{18}O$ at DF (Fig. 3) and displays high and low values during interglacials and cold periods in glacials, respectively. Similarly, residual $nssSO_4^{2-}$ fluxes calculated for EDC and EDML show variability consistent with DF (Fig. 3). Opposite behaviors of marine biogenic and terrestrial sulphate would have led to small variability of the $nssSO_4^{2-}$ flux over glacial/interglacial cycles. Larger dust input at DF and EDML owing to their proximity to the South American source regions relative to EDC (Fig. 1b, Supplementary Fig. 3a) would have resulted in greater variability in the $nssSO_4^{2-}$ flux at DF and EDML compared with EDC (Fig. 1c, Supplementary Fig. 3b).

As stated above, we first subtract the terrestrial gypsum contribution to calculate the marine biogenic $nssSO_4^{2-}$ flux. However, because $nssSO_4^{2-}$ could have other sources, we perform a sensitivity test as follows. If a major fraction of $nssSO_4^{2-}$ originates from evaporites, then other minerals commonly contained in evaporites could also contribute to the $nssSO_4^{2-}$ flux. We take into account $Mg^{2+}$ and $K^+$, which could originate from evaporites and exist as sulphates[29], as well as the contribution of $CaCO_3$, a major mineral in many of the dust source regions[26,27] that likely reacts with $HNO_3$ or $NO_x$ instead of $H_2SO_4$ or $SO_2$ due to faster reactions[20,25,32,33,36] (Supplementary Discussion and Supplementary Fig. 4). Our conclusion that the residual $nssSO_4^{2-}$ flux decreases during cold periods does not change, although the correlation between residual sulphate and temperature proxy changes slightly (Fig. 4a, b, Supplementary Fig. 5). We also change the $nssSO_4^{2-}/nssCa^{2+}$ ratio ($R_1$) assuming that part of $CaSO_4$ originates from the reaction of $CaCO_3$ with marine biogenic sulphate. When we change $R_1$ values, we consider only $CaSO_4$ and ignore other minerals. The same conclusion remains if $R_1 > 1.2$, but fails if $R_1 < 1.2$. For EDC and EDML cores, we consider only the $CaSO_4$ contribution because neither $Mg^{2+}$ nor $K^+$ data are available.

We calculate the marine biogenic/total nss $SO_4^{2-}$ ratio ($R_2$) for different terrestrial gypsum contributions (Supplementary Discussion and Supplementary Fig. 6). If we assume that $Ca^{2+}$ and $Mg^{2+}$ are major evaporite-originated cations that form sulphate in DF core and that the carbonate hosts of these ions react with $HNO_3$ or $NO_x$ rather than $H_2SO_4$ or $SO_2$, $R_2$ for the LGM is 0.46. This value is consistent with that estimated from sulphur isotopes

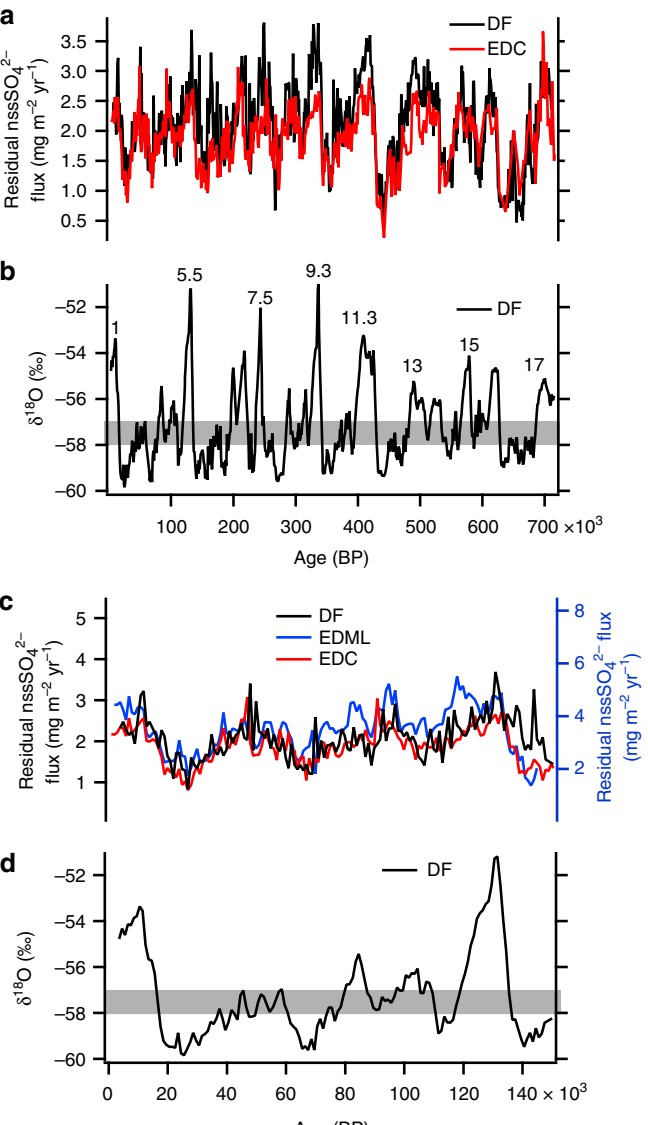

**Fig. 3** Variability of residual $nssSO_4^{2-}$ flux at Dome Fuji (DF) (EDC), and Dronning Maud Land (EDML). **a** Residual $nssSO_4^{2-}$ flux at DF and EDC for the past 720,000 years, calculated by subtracting the terrestrial $CaSO_4$ contribution from the $nssSO_4^{2-}$ flux. The EDC flux is plotted on the AICC12 timescale[53,54] using previously published ion data[7-9,16] and accumulation rates[53,54]. **b** $\delta^{18}O$ at DF over the past 720,000 years[14,15]. Gray bar indicates the thresholds (Fig. 2a). Marine isotope stage numbers for interglacials are also shown. **c** Residual $nssSO_4^{2-}$ at DF, EDC, and EDML calculated by subtracting the terrestrial $CaSO_4$ contribution from the $nssSO_4^{2-}$ flux. The EDC and EDML fluxes are plotted on the AICC12 timescale[53,54] using previously published ion data[7-9,16] and accumulation rates[53,54]. **d** The $\delta^{18}O$ values at DF[14] for the past 150,000 years. All ion and $\delta^{18}O$ values are averages over 1000 years

($R_2 \sim 0.5$) assuming no isotopic fractionation[11]. However, $R_2$ values for interglacials exceed 1, which is implausible. This most likely suggests an overestimation of the $NO_3^-$ derived from the reaction between carbonates and $HNO_3$ or $NO_x$, because $NO_3^-$ can also exist as $HNO_3$. If we consider only terrestrial gypsum as a major contributor to terrestrial $nssSO_4^{2-}$, $R_1 = 2.4$, 1.5, and 1.3 yield $R_2 = 0.24$, 0.52, and 0.59, respectively (Supplementary Fig. 6) for the LGM. The same $R_1$ values yield $R_2 = 0.36$, 0.60, and 0.65, respectively, for EDC core, and $R_2 = 0.24$, 0.56, and 0.60, respectively, for EDML core. Larger dust input at DF and

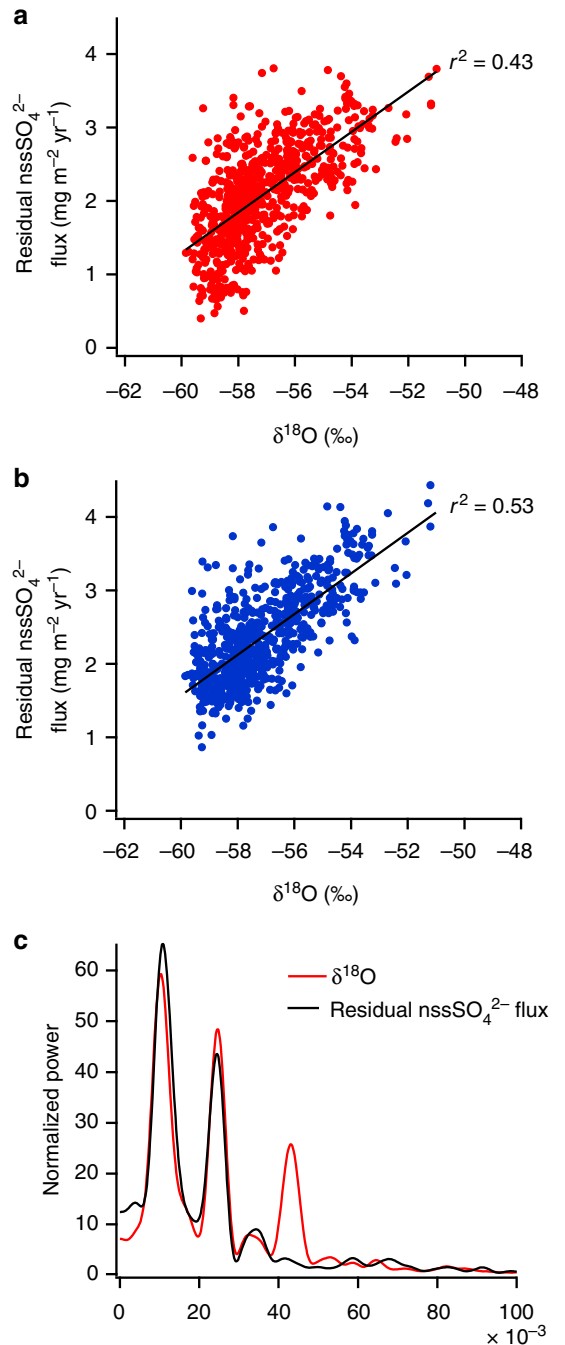

EDML likely yields smaller $R_2$ values compared with EDC. $R_2$ values for the LGM might be underestimated for large $R_1$ values (~2.4) owing to contribution of marine biogenic sulphate. In any case, a contribution of terrestrial sulphate during glacials is likely much larger than previously assumed, consistent with that estimated from sulphur isotope mass balance[11]. This leads to a conclusion that DMS-derived sulphate decreases in glacials, although the degree of decrease remains uncertain depending on glacial/interglacial changes in $R_2$ values.

**Reduced DMS emissions in the AZ of the SO during glacials.** Sulphate aerosol observations at EDC and coastal Antarctic sites display a clear seasonal pattern with a maximum in austral summer[37]. High surface DMS concentrations and emission fluxes over the modern SO in austral summer have also been reported[38,39]. The dominant source of biogenic sulphate in the Antarctic interior is thus most likely DMS emitted from the SO. The flux of marine biogenic sulphate deposited in the Antarctic interior would then be controlled by DMS emissions in the source regions, the location of these source regions (i.e., distance to Antarctic interior sites), DMS oxidation chemistry, and depositional processes. As glacial/interglacial changes in oxidation chemistry and deposition are likely to be small, the decreased biogenic sulphate flux during glacials would be caused by reduced DMS emissions and/or longer transport distances[40] (Supplementary Discussion). Transport distances depend strongly on the summer sea ice extent around Antarctica[40] (Supplementary Discussion), but only limited information is available for glacials. In the Indian Ocean sector, the summer sea ice extent at the LGM was only slightly greater than the present day[41], whereas in the Atlantic Ocean sector, the sporadic occurrence of summer sea ice considerably farther north is indicated[41] (Supplementary Discussion). Although data from other oceanic sectors are very sparse, Gersonde et al.[41] speculate that the summer sea ice field around Antarctica changed from $4 \times 10^6$ km$^2$ (present day) to $5–6 \times 10^6$ km$^2$ (LGM).

DF and EDC display similar residual nssSO$_4^{2-}$ fluxes with glacial/interglacial ratios of 1/3 to 1/4 (Fig. 3). The LGM/Holocene ratios (~1/3) at DF, EDC, and EDML are consistent with the MSA flux at Siple Dome (West Antarctica, Supplementary Fig. 1) where MSA can be used as a proxy for marine biogenic sulphur deposition because its post-depositional loss is minimal[42]. In other words, the four sites facing different sectors of the SO, which includes the Indian sector where summer sea ice extent increased only slightly at the LGM, display similar glacial/interglacial ratios of biogenic sulphur species. The similar ratios would be mainly associated with glacial-interglacial changes in DMS emissions in the SO, and specifically the AZ because it is a major DMS source region for biogenic sulphate in the present-day Antarctic interior[38,43] (Supplementary Discussion). The reduced biogenic sulphate fluxes at the LGM could be partly due to the increased transport distances. However, the LGM increase in the summer sea ice extent around Antarctica by 1.25 to 1.50 times[41] would only slightly increase the transport distances to the Antarctic interior sites, which are affected by a mixture of air masses from different oceanic sectors[44,45] (Supplementary Discussion and Supplementary Fig. 7). Thus, lower biogenic nssSO$_4^{2-}$ fluxes during glacials indicate reduced DMS emissions in the AZ, suggesting that primary production, as well as export production, decreases during glacials, which is consistent with marine sediment records[4].

**Discussion**
Sea surface temperature (SST), solar radiation, sea ice extent, and nutrient and iron supply can affect DMS emissions[1,43,46,47] in the

**Fig. 4** Relationship between residual nssSO$_4^{2-}$ flux and δ$^{18}$O at Dome Fuji (DF). **a** Residual nssSO$_4^{2-}$ flux, considering only the terrestrial CaSO$_4$ contribution, plotted against δ$^{18}$O[14,15]. **b** Residual nssSO$_4^{2-}$ flux considering the contributions of CaSO$_4$, MgSO$_4$, Ca(NO$_3$)$_2$, and Mg(NO$_3$)$_2$ plotted against δ$^{18}$O[14,15]. Residual nssSO$_4^{2-}$ flux and δ$^{18}$O are averages over 1000 years. Before averaging, the δ$^{18}$O depths that differ from the ion data depths have been interpolated to match. Straight lines in **a** and **b** display results of linear regressions. Correlation coefficients ($r$) were calculated with sample size ($n$) = 681 and for significance level ($\alpha$) = 0.05. **c** Normalized power spectra of residual nssSO$_4^{2-}$ flux and δ$^{18}$O at DF. The residual nssSO$_4^{2-}$ flux was calculated in the same manner as **b**. Power spectra were calculated with the Blackman-Tukey method (30% lag) using the Analyseries software package[55] (see Methods). To use the software, the raw data were resampled to a 200-yr interval using linear interpolation

AZ. Power spectra of residual $nssSO_4^{2-}$ show strong powers in the 41-kyr and 93-kyr bands (Fig. 4c). Powers in similar bands (41-kyr and 98-kyr) are also observed in the $\delta^{18}O$ record, which is closely linked to SST and sea ice extent in the AZ. To our knowledge, the relationship between DMS emissions and SST has not been directly investigated. The growth rate of phytoplankton (unicellular algae), however, shows little dependence on SST near the melting point of sea ice[48], which is the major source of DMS[46,49]. Hence, covariance of the residual $nssSO_4^{2-}$ flux and $\delta^{18}O$ record at DF (Figs. 3, 4, Supplementary Fig. 5) does not imply that decreased summer SST is a major cause of reduced DMS emissions. Although the integrated summer insolation at 55°S, the latitude of a major source region of DMS, shows strong spectral power in the 41-kyr band, variability in solar radiation could not be a major cause of the reduced DMS emissions during glacials because it is less than 3% (Supplementary Discussion). The large seasonal difference in sea ice extent during glacials implies large areas of melting sea ice in summer, which would lead to enhanced DMS emissions because melting sea ice is an important DMS source[46,49]. However, this is not the case because DMS-derived sulphate decreases in glacials (Figs. 3, 4, Supplementary Fig. 5). Thus, the change in winter sea ice extent does not directly affect overall DMS emissions in the AZ on orbital timescales.

Vertical mixing and upwelling appear to dominate the nutrient and iron supply in Antarctic surface waters[4]. Expanded winter sea ice during glacials would enhance AZ stratification, weaken mixing and upwelling, and decrease the supply of nutrients and iron in winter[4]. This would decrease the nutrient/iron abundance and thus DMS emissions in summer. Reduced vertical mixing and upwelling during glacials should also reduce the $CO_2$ exchange between the ocean interior and atmosphere, thereby sequestering $CO_2$ into the ocean and leading to decreased atmospheric $CO_2$ concentrations, as is proposed by[4]. This study also implies that reduced DMS emissions during glacials may reduce cloud albedo, resulting in a negative feedback by biogenic sulphate aerosol-cloud interaction[1,2]. Although an improved understanding of the precise mechanisms controlling $nssSO_4^{2-}$ flux variations and their links to climate change is needed, the data provided here can be used to constrain the sulphur cycle and climate models. Ongoing analyses of sulphur isotopes of $SO_4^{2-}$ ($\delta^{34}S$) in DF core will reduce the estimation uncertainty of DMS-derived sulphate, and enable more quantitative discussion on the interaction between DMS-derived sulphate and climate.

## Methods

**Ion data**. We use ion data from DF1 and DF2 cores after and before 300,000 BP, respectively[14]. $Na^+$, $Ca^{2+}$, $Mg^{2+}$, $NO_3^-$, and $SO_4^{2-}$ were measured from both cores using ion chromatography. In addition, $K^+$ was measured from DF2 core. For DF1 core, we use previously published data[50] after re-examination and removal of some data points because of large measurement errors. Fifty-nine samples were newly cut from DF1 core, re-measured, and the new data were added to the earlier dataset. Measurement errors were generally less than 10% but may be higher for low concentrations. For DF2 core, 10-cm-long samples were cut every 0.5 m and measured on two Dionex DX-500 ion chromatographs: one for anions and the other for cations. Measurement errors were estimated to be less than 3%. Sea salt (ss) $Na^+$ and non-sea-salt (nss) $Ca^{2+}$ concentrations were calculated from $Na^+$ and $Ca^{2+}$ concentration data using the weight ratios of $Ca^{2+}/Na^+$ for seawater (0.038) and average crust (1.78), as described in previous studies[7–9,16,51]. The $nssSO_4^{2-}$ concentrations were calculated assuming a sea ice source[7,8] of $ssNa^+$. Similar values are obtained if we assume an open ocean source[7,8] of $ssNa^+$. Fluxes of $nssCa^{2+}$ and $nssSO_4^{2-}$ were calculated by multiplying concentrations by estimated accumulation rates[14].

**Chronology and accumulation rate estimation**. We use the DFO-2006[52] timescale for the past ~342,000 years and the AICC2012[53] timescale for the period older than ~344,000 years[14]. The AICC2012[53,54] chronology is used for EDC and EDML. The accumulation rates at DF were deduced from the $\delta^{18}O$ record by Dome Fuji Community members[14], and those at EDC and EDML were taken from [53,54].

**Spectral analysis**. Spectral analyses were carried out with the Analyseries software package[55] (Fig. 4c). Blackman-Tukey spectra (30% lag) using a Bartlett window with a bandwidth of 0.00702905 are shown in Fig. 4c. The amplitudes of the spectra were normalized. The $\delta^{18}O$ and residual $nssSO_4^{2-}$ data used for the spectral analysis were resampled at a 200-yr interval using linear interpolation. For resampling, $\delta^{18}O$ data from[14] and residual $nssSO_4^{2-}$ data provided in the Source Data file were used.

## Data availability

The source data underlying Figs. 1–4 and Supplementary Figs. 2–7 are provided as a Source Data file. The data are also available in the Arctic and Antarctic Data Archive System at the National Institute of Polar Research [https://ads.nipr.ac.jp/dataset/A20190607-001].

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

## Acknowledgements

We thank the Dome Fuji Deep Ice Core Project members who contributed to the ice coring and ice core processing. The Japanese Antarctic Research Expedition (JARE), managed by the Ministry of Education, Culture, Sports, Science and Technology (MEXT), provided primary logistical support for the Dome Fuji Deep Ice Core Project. We thank Eric Wolff and Hubertus Fischer who provided the EDC and EDML ion data and valuable comments on the manuscript. We thank Yutaka Tobo for discussions regarding the chemical reactions of mineral aerosols, and two reviewers for their constructive comments. We also thank Yoshimi Ogawa-Tsukagawa for her help in drawing diagrams. This study was supported in part by MEXT (Grant-in-Aid for Scientific Research 15101001, 21221002, 15H01731, and 17H06316) and National Institute of Polar Research (Project Research KP305).

## Author contributions

H.M. and Y.F. recovered the Dome Fuji cores. K.G.-A., M.H., T.M., T.K., R.U., K.S., Y.I, T. Suzuki, S. Horikawa, and K.F. processed the cores. M.H. and M.I. performed ion analyses. R.U. and H.M. obtained stable isotope data. T.K., M.H., and K.G.-A. were responsible for quality control of the ion data. K.G.-A. led the manuscript preparation. R.U., T. Suzuki, T. Sakurai, and K.F. contributed to the discussion on the ice core data. Y.K. contributed to the discussion on DMS and overall manuscript preparation. S. Hattori contributed to the discussion on sulphur isotopes.

## Additional information

**Competing interests:** The authors declare no competing interests.

