## [Peer Review File · Nature Communications]

Reviewer #1 (Remarks to the Author):

This paper addresses a finding from the EPICA ice cores that was surprising – namely that the non-sea-salt sulfate flux to Antarctica was constant (within ~15%) through glacial and interglacials (most clearly stated in Wolff et al 2006 and 2010, and Kaufmann et al 2010). The apparently constant flux was important because it refuted the idea that significantly increased marine sulfur-emitting productivity occurred in glacial and acted as a positive feedback on climate. That idea was persistent in the literature even though it had become apparent that it was based itself on MSA data that turned out to be a postdepositional artefact. It was not clear whether the quasi-constancy arose because productivity of the relevant organisms was unaffected by glacial cycles or because of a coincidence of factors cancelling each other out. In the current paper the authors re-address this issue, using data from Dome Fuji. They propose the idea (already mentioned but not followed up in Wolff et al 2010) that terrestrial sulfate may be important, and that, when corrected for, it suggests lower marine productivity in glacial, in better agreement with marine sediment data.

With the new data from Dome Fuji, I do believe that this topic deserves a careful re-examination and may eventually be suitable for Nature Communications. However the authors are way too partisan here, pushing their theory in the face of several lines of evidence that make it (at least in the form they have proposed) very unlikely. A new version in which they clearly lay out the arguments for and against their idea is required, with a likely outcome that at this stage we cannot decide between the hypotheses – that terrestrial input is or is not important, with their opposite implications for the marine productivity story. I will lay out here the main issues that need to be discussed.

Firstly it is interesting but not decisive that DF nss-S (shorthand for nss-sulfate for the purpose of this review) shows a glacial-interglacial pattern while EDC and EDML do not. Firstly, it is by no means clear that there is a major glacial cycle in nss-S at DF. With the current presentation in Fig 1c, it is rather hard to tell, because one's eye is drawn to the upper limits of the noise rather than to the smoothed signal. Looking at the older glacial cycles, and at the quite small dynamic range of Fig 2a, there seems like only a rather small contrast between glacial and interglacial. The data should be replotted with comparable time resolution (eg 1 kyr) all along the core so that the real pattern can be discerned. My feeling is that, when this is done, there will be only a subtle contrast between sites: as an example, Fig S3b suggests that both EDML and DF show 50-100% increases in nss-S just during the LGM (a period of about 15 kyr). I therefore recommend a more nuanced discussion showing that the glacial-interglacial contrast is there, but quite small, and not that dissimilar between the two sites facing the same sector of ocean.

It should also be emphasised that there is no a priori reason why we should get the same result for the Indian and Atlantic Ocean sectors, nor is it obvious what result marine records suggest. Decreased productivity at the LGM south of the Antarctic Polar Front contrasts with increased productivity to the north, so that an inland Antarctic site might see any combination of those two influences. The paper cited in the SI (Cosme et al, lines 116-123 of SI) specifically suggests a significant fraction of S originates north of the APF, so while the authors are correct to imply that S from south of the APF may be a majority, it is misleading to call it dominant. Finally, and I know that the authors are aware of this, it is not obvious that the productivity of the S-producers and the productivity measured in marine cores are synonymous – but that is just another complication that has to be borne in mind.

The prime argument the authors use for subtracting a terrestrial component is the observed relationship between Ca and sulfate in Fig 2b. This is indeed interesting, but I do not read it the way the authors do. The ratio of the lower envelope is at the stoichiometric ratio of Ca:sulfate but with a positive intercept. This indeed might be taken as evidence that sulfate and Ca are being deposited together but this can arise from two possibilities: (a) that gypsum is being transported

from a terrestrial source or (b) that calcium salts are reacting in the atmosphere with sulfate and then being deposited more efficiently than sulfuric acid. The authors go for option (a) but this has several flaws:

1. The plot (Fig 2b), if taken as the authors suggest, implies that ALL Ca is present as gypsum. This would imply an incredibly unlikely source that has gypsum and no other Ca salts. The ratio of sulfate to calcium implied (2.4) is about 20 times that of the average crust and much higher than that of even sulfate rich source areas. To support their interpretation the authors would need to demonstrate significant source regions where gypsum was the main mineral available for transport, which I don't think they do.

2. Further to that, the authors attempt later to suggest that $MgSO_4$ could also be a significant player. However, if they include Mg (apparently correlated with Mg and at about 35% of its concentration) then they undermine their argument that there is some significance to the stoichiometric ratio of the line in Fig 2b: if 35% of the terrestrial sulfate is attached to Mg, then the slope with Ca should be at 65% of the stoichiometric ratio (I appreciate that 35 may not be the right number here but hope the point is clear)

There is one piece of evidence the authors don't quote which might partly support their case and that is S isotope evidence from Alexander et al (2003). This shows (with very sparse data) more negative values in the glacial. While Alexander et al gave an explanation based on fractionation during transport, this was partly because they dismissed a major continental influence because they had in their heads a paradigm of increased marine influence in glacials. However the isotope data could be taken as evidence for an increased continental input during glacials, though I have not done the calculations to see what level of input would be required to explain their data.

Another argument the authors imply supports their case is that, once the "correction" is applied, the residual δS is similar at the three locations. However this is no argument: we know that Ca is highly correlated at the three sites with vastly higher Ca fluxes in glacials. If you subtract something with the Ca pattern from something without much variability then it is bound to end up with a pattern opposite to that of Ca, ie it will be low in glacials, and the same at all sites. This is mathematically obvious and cannot be used to support making the correction.

The result of all this is that the authors need to be more open to two alternative possibilities, neither of them completely supported:

1. The correction the authors have done might be partly supported by S isotopes, but relies on an extremely surprising source ratio where gypsum is the ONLY Ca salt present, and one which becomes even more strange if the Mg correction is done as well.

2. As an alternative the $CaSO_4$ might arise from reactions on dust particles in the atmosphere. It would also be justified to correct for the more efficient deposition of such dust in comparison to sulfuric acid that has not reacted or that reacts with sea salt, but it is not immediately obvious that a 1:1 correction would be justified. One argument (I admit) against this is that it requires dust and sulfate to be in the atmosphere in the same season, and this is a weak point, perhaps arguing for a combination of 1 and 2.

If the authors make this more nuanced explanation they will have to admit to less certainty about the strength of the correction and to less certainty about the change in S production across glacial cycles. However they will have carefully explained the issues and made an important contribution suggesting further studies (for example, more S isotope work is clearly called for). I therefore encourage resubmission but only after major revision.

A few more detailed comments:

Abstract, line 1. "is a measure of primary productivity". Given that only a subset of phytoplankton are S producers, "may be an indicator of primary..." would be safer. Same in line 52 on next page.

Lines 56/57. It would be fair here to add "but increased productivity further north".

Line 88 and Fig 1. Please give a clearer description (e.g. "with values x% higher in the LGM than the Holocene") and show the data with a constant smoothing so we can see the underlying trend. If not already done the data in Fig 2 should also be with constant resolution to avoid noisy outliers.

Line 116. It is a huge jump from saying that sometimes CaSO₄ is observed near sources to saying "It is reasonable to assume that continental gypsum is dominant". Please discuss the fact that your correction implies that ALL Ca is present as gypsum which seems very strange.

Line 125 and following. Please make clear that it is inevitable that subtracting a multiple of Ca (even a much smaller one than you use) from a flattish record will inevitably lead to the pattern observed and to a similar pattern at each site so this should not be used as evidence supporting your decision. Thus for example in line 129 "This suggests that terrestrial CaSO₄....during glacials" is not really a correct inference.

Line 137 paragraph. This is very confusing and doesn't quite make sense until one reads the SI. I suggest a shorter paragraph in which you say that one could also correct for other salts (see SI) as a sensitivity study without greatly affecting the result, but that this would undermine the argument about the Ca/SO₄ ratio.

However please also discuss the implications if a large part of the Ca:SO₄ correlation does arise from atmospheric reactions and how this would affect the correction you make.

SI, line 52. I do not follow how you make this conclusion that "primary is ...major" and secondary is minor. By all means there may be a reaction with nitrate but it is not clear why this rules out a reaction with sulfate.

SI, section on MSA. I am deeply unconvinced by this calculation. Your calculation multiplying MSA by 1/A seems entirely unjustified – there is no mechanistic reason why this should be the case, nor for the choice of del18O threshold (which mechanistically should be a threshold in dust concentration). Of course with suitable tuning of A one can reproduce higher values in interglacials but this is entirely fabricated and seems to have no validity. I urge scrapping this section.

SI, lines 117-121. Please see my comments above on the relative roles of AZ and SAZ.

Line 121. Surely the APF is the northern edge of the AZ not southern?

Reviewer #3 (Remarks to the Author):

Review of Goto-Azuma et al, 'Reduced marine phytoplankton sulphur emissions in the Southern Ocean during the past seven glacial periods', submission to Nature Communications, July 2018.

This paper presents new ionic composition data of the Dome Fuji ice core, along with published data from the EPICA ice cores, to suggest there were glacial interglacial changes in 'residual non-seasalt-sulfate' which is interpreted to be related to biogenic sulfur emissions from the Southern Ocean. There is higher residual non-seasalt sulfate during interglacial periods than glacial periods and this is related to support other observations of higher marine productivity in the Antarctic zone during interglacials.

I think this manuscript is very interesting and understanding the connection between the marine sulfur cycle and global climate change is very important to thinking in the field. I think there needs to be more justification of exactly how the authors chose to calculate the residual non-seasalt sulfate before publication could be considered, as this differs from the previous studies that they cite of the EDC core.

In particular, the Kaufman et al. (2010, QSR) study cited evaluates how much of the sulfate could be contributed by mineral dust. In the Kaufman study, they assumed a maximum sulfate/calcium ratio of dust of 0.5 (mass ratio) based on measurements of Saharan dust (agreeably potentially not the most relevant material) and calculated a maximum of 16% mineral sulfate. The present paper assumed sulfate/calcium of dust = 2.4, almost 5 times higher, and this results in what looks like 75% mineral sulfate in some cases (hard to tell without having the tabulated data). In other words, the present paper is assuming all of the Ca in mineral dust is CaSO_4 . The authors deduce that any CaCO_3 originally in the dust would have been converted to $\text{Ca(NO}_3)_2$ rather than sulfate. Fair enough. But what if the majority of the Ca is as silicate minerals (clays, plagioclase, feldspar, etc) which contain Ca but no SO_4 ? The Sakurai et al. study they cite as determining all the Ca^{2+} is as gypsum also seems to measure a substantial fraction of feldspar (as 'other'), although that particular method may not be the best for determining calcium silicates. I think more justification needs to be given as to the mineralogical composition of the Patagonian dust and it should be clearly stated why the SO_4/Ca ratio they use is different from that which was assumed previously. One other study I came across while doing this review may be of use. Alexander et al. (2003, JGR doi: 10.1029/2003JD003513) measured sulfur isotopes in Vostok and Dome C sulfate. This study found that the isotope composition of the sulfate potentially implied a large proportion of mineral sulfate (~50%). However, because an estimate of mineral sulfate using Ca, Na, Mg and NO_3 , citing LeGrand et al 1988 Atmos Environ) suggested <10% mineral sulfate, these authors concluded that the isotope anomaly must therefore be reflecting a fractionation process rather than simple end-member mixing. Perhaps this study would need to be reinterpreted as well if the mineral sulfate really is as high as the present authors suggest.

If this major issue can be settled, the subsequent discussion of the possible causes and/or consequences of varying marine sulfur emissions are very valuable targets for future global climate research.

Responses to Reviewers' comments

We thank the reviewers for their important and constructive comments. Our response to each comment is written under it in blue italic characters.

Reviewer #1 (Remarks to the Author):

This paper addresses a finding from the EPICA ice cores that was surprising ? namely that the non-sea-salt sulfate flux to Antarctica was constant (within ~15%) through glacial and interglacials (most clearly stated in Wolff et al 2006 and 2010, and Kaufmann et al 2010). The apparently constant flux was important because it refuted the idea that significantly increased marine sulfur-emitting productivity occurred in glacial and acted as a positive feedback on climate. That idea was persistent in the literature even though it had become apparent that it was based itself on MSA data that turned out to be a postdepositional artefact. It was not clear whether the quasi-constancy arose because productivity of the relevant organisms was unaffected by glacial cycles or because of a coincidence of factors cancelling each other out. In the current paper the authors re-address this issue, using data from Dome Fuji. They propose the idea (already mentioned but not followed up in Wolff et al 2010) that terrestrial sulfate may be important, and that, when corrected for, it suggests lower marine productivity in glacial, in better agreement with marine sediment data.

We also think that the findings of Wolff et al. (2006 and 2010) and Kaufmann et al. (2010) are very important because they refuted the prevailing idea that significantly increased marine sulfur-emitting productivity occurred in glacial and acted as a positive feedback on climate. In Introduction, we have explained how interpretations of ice core sulphate and MSA data have changed over the last three decades.

With the new data from Dome Fuji, I do believe that this topic deserves a careful re-examination and may eventually be suitable for Nature Communications. However the authors are way too partisan here, pushing their theory in the face of several lines of evidence that make it (at least in the form they have proposed) very unlikely. A new version in which they clearly lay out the arguments for and against their idea is required, with a likely outcome that at this stage we cannot decide between the hypotheses ? that terrestrial input is or is not important, with their opposite implications for the marine productivity story. I will lay out here the main issues that need to be discussed.

We still think that terrestrial input is important, based on a new interpretation of the sulphur isotopic composition study by Alexander et al. (2003), as was suggested by Reviewers #1 and #3. When Alexander et al. published the paper, they had to assume isotopic fractionation to explain 4-6‰ lower $\delta^{34}\text{S}$ for the last glacial with respect to the Holocene, because they considered terrestrial

input was low. However, a recent study on surface snow samples from a latitudinal transect between a coastal station (Syowa) and interior site (Dome Fuji) suggests that isotopic fractionation during long-range transport is insignificant in East Antarctica and thus $\delta^{34}\text{S}$ in the ice cores from the East Antarctic interior can be used to infer source contributions (Uemura et al.2016). We have explained this in Introduction (Lines 82-93) and Results (Lines 203-204).

As Reviewer #1 commented, there are large uncertainties. We have changed some of the sentences so that they do not sound decisive. As English is not our mother tongue, it is not so easy to change the English of our manuscript properly. Although we used an English editing service, it did not work very well. Some of the sentences may still give an impression that we are pushing our theory. We would appreciate suggestions from Reviewer #1 how we could change them.

Firstly it is interesting but not decisive that DF nss-S (shorthand for nss-sulfate for the purpose of this review) shows a glacial-interglacial pattern while EDC and EDML do not. Firstly, it is by no means clear that there is a major glacial cycle in nss-S at DF. With the current presentation in Fig 1c, it is rather hard to tell, because one's eye is drawn to the upper limits of the noise rather than to the smoothed signal. Looking at the older glacial cycles, and at the quite small dynamic range of Fig 2a, there seems like only a rather small contrast between glacial and interglacial. The data should be replotted with comparable time resolution (eg 1 kyr) all along the core so that the real pattern can be discerned. My feeling is that, when this is done, there will be only a subtle contrast between sites: as an example, Fig S3b suggests that both EDML and DF show 50-100% increases in nss-S just during the LGM (a period of about 15 kyr). I therefore recommend a more nuanced discussion showing that the glacial-interglacial contrast is there, but quite small, and not that dissimilar between the two sites facing the same sector of ocean.

We do not intend to say that nss sulphate shows a clear glacial-interglacial pattern, because it is a complicated pattern as seen in Fig.2a. We would just like to point out that nss sulphate is not constant at DF, and changed the text (Lines 114-117). All the data are replotted with 1kyr resolution. The suggestion to replot the data was very helpful. Now the figures show the variability in nss sulphate better. Though EDC shows very small variability, both DF and EDML show high nss sulphate fluxes during cold periods. We think this is because higher dust input at DF and EDML during cold periods (Lines 181-185).

It should also be emphasised that there is no a priori reason why we should get the same result for the Indian and Atlantic Ocean sectors, nor is it obvious what result marine records suggest. Decreased productivity at the LGM south of the Antarctic Polar Front contrasts with increased productivity to the north, so that an inland Antarctic site might see any combination of those two influences. The paper cited in the SI (Cosme et al, lines 116-123 of SI) specifically suggests a significant fraction of S originates north of the APF, so while the authors are

correct to imply that S from south of the APF may be a majority, it is misleading to call it dominant. Finally, and I know that the authors are aware of this, it is not obvious that the productivity of the S-producers and the productivity measured in marine cores are synonymous ? but that is just another complication that has to be borne in mind.

We agree that there is no a priori reason why we should get the same result for the Indian and Atlantic Ocean sectors, and have changed the text (Lines 177-182). We have changed the word “dominant” to “major” (SI Line 94). Although we agree that the S-producers and the productivity measured in marine cores are not synonymous, we did not discuss this in the manuscript, because it would make the discussion more complicated. We just tried not to use the marine core results as evidence of higher S productivity in glacials.

The prime argument the authors use for subtracting a terrestrial component is the observed relationship between Ca and sulfate in Fig 2b. This is indeed interesting, but I do not read it the way the authors do. The ratio of the lower envelope is at the stoichiometric ratio of Ca:sulfate but with a positive intercept. This indeed might be taken as evidence that sulfate and Ca are being deposited together but this can arise from two possibilities: (a) that gypsum is being transported from a terrestrial source or (b) that calcium salts are reacting in the atmosphere with sulfate and then being deposited more efficiently than sulfuric acid. The authors go for option (a) but this has several flaws:

1. The plot (Fig 2b), if taken as the authors suggest, implies that ALL Ca is present as gypsum. This would imply an incredibly unlikely source that has gypsum and no other Ca salts. The ratio of sulfate to calcium implied (2.4) is about 20 times that of the average crust and much higher than that of even sulfate rich source areas. To support their interpretation the authors would need to demonstrate significant source regions where gypsum was the main mineral available for transport, which I don't think they do.

We have demonstrated significant source regions where gypsum is a major mineral (Lines 153-163). We also think that CaCO_3 is contained in South American source regions. We assumed that majority of CaCO_3 reacted with HNO_3/NO_x rather than $\text{H}_2\text{SO}_4/\text{SO}_2$ and explained the reason for this assumption (Lines 163-165, 191-193, SI Lines 56-69). But we also performed sensitivity tests using different ratios of sulphate to calcium (Lines 199-213).

2. Further to that, the authors attempt later to suggest that MgSO_4 could also be a significant player. However, if they include Mg (apparently correlated with Mg and at about 35% of its concentration) then they undermine their argument that there is some significance to the stoichiometric ratio of the line in Fig 2b: if 35% of the terrestrial sulfate is attached to Mg, then the slope with Ca should be at 65% of the stoichiometric ratio (I appreciate that 35 may not be the right number here but hope the point is clear)

For a first-order approximation, we considered only CaSO₄. For a better approximation, we should also consider MgSO₄, Ca(NO₃)₂ and Mg(NO₃)₂. Detailed explanation is given in SI (SI Lines 42-85 and Fig. S5).

There is one piece of evidence the authors don't quote which might partly support their case and that is S isotope evidence from Alexander et al (2003). This shows (with very sparse data) more negative values in the glacial. While Alexander et al gave an explanation based on fractionation during transport, this was partly because they dismissed a major continental influence because they had in their heads a paradigm of increased marine influence in glacials. However the isotope data could be taken as evidence for an increased continental input during glacials, though I have not done the calculations to see what level of input would be required to explain their data.

We appreciate this comment very much. Alexander et al. (2003) supports our hypothesis. If they assume no fractionation during transport, the continental source represents ~50% of the nss sulphate in their samples. This ratio agrees well with that estimated from our residual nss sulphate. We have explained this in the text (Lines 199-213) and added Fig. S6.

Another argument the authors imply supports their case is that, once the "correction" is applied, the residual nss-S is similar at the three locations. However this is no argument: we know that Ca is highly correlated at the three sites with vastly higher Ca fluxes in glacials. If you subtract something with the Ca pattern from something without much variability then it is bound to end up with a pattern opposite to that of Ca, ie it will be low in glacials, and the same at all sites. This is mathematically obvious and cannot be used to support making the correction.

We have removed this argument.

The result of all this is that the authors need to be more open to two alternative possibilities, neither of them completely supported:

1. The correction the authors have done might be partly supported by S isotopes, but relies on an extremely surprising source ratio where gypsum is the ONLY Ca salt present, and one which becomes even more strange if the Mg correction is done as well.

As written above, we have demonstrated significant source regions where gypsum is a major mineral. This implies that source ratio could be much higher than previously believed. Although we have shown two alternative possibilities, we believe that the contribution of terrestrial gypsum (and MgSO₄ as well) was much greater than previously assumed.

2. As an alternative the CaSO₄ might arise from reactions on dust particles in the atmosphere. It would also be justified to correct for the more efficient

deposition of such dust in comparison to sulfuric acid that has not reacted or that reacts with sea salt, but it is not immediately obvious that a 1:1 correction would be justified. One argument (I admit) against this is that it requires dust and sulfate to be in the atmosphere in the same season, and this is a weak point, perhaps arguing for a combination of 1 and 2.

We do not say that no CaSO₄ arises from reactions on dust particles in the atmosphere. But according to the literatures, the reactions with H₂SO₄/SO₂ are slow (Lines 163-165). We have therefore assumed that the contribution of CaSO₄ originated from reactions on dust was minor and that there was significant contribution from terrestrial sulphate (CaSO₄ and MgSO₄) in cold periods.

If the authors make this more nuanced explanation they will have to admit to less certainty about the strength of the correction and to less certainty about the change in S production across glacial cycles. However they will have carefully explained the issues and made an important contribution suggesting further studies (for example, more S isotope work is clearly called for). I therefore encourage resubmission but only after major revision.

We have tried to make more nuanced explanation compared to the first version of the manuscript. We have carried out sensitivity tests as written above. We have also written about further sulphur isotope work (Lines 276-278).

A few more detailed comments:

Abstract, line 1. "is a measure of primary productivity". Given that only a subset of phytoplankton are S producers, "may be an indicator of primary..." would be safer. Same in line 52 on next page.

We have changed the sentence as suggested.

Lines 56/57. It would be fair here to add "but increased productivity further north".

We have added this.

Line 88 and Fig 1. Please give a clearer description (e.g. "with values x% higher in the LGM than the Holocene") and show the data with a constant smoothing so we can see the underlying trend. If not already done the data in Fig 2 should also be with constant resolution to avoid noisy outliers.

We have removed the sentence (Line 88 of the first version of the manuscript) "Contrary to the previous studies on the EDC and EDML cores, the nssSO₄²⁻ flux at DF displays climate-dependent variations. We have replotted Fig.2 (and all other figures) with 1 kyr constant resolution.

Line 116. It is a huge jump from saying that sometimes CaSO₄ is observed near sources to saying “It is reasonable to assume that continental gypsum is dominant”. Please discuss the fact that your correction implies that ALL Ca is present as gypsum which seems very strange.

We agree with the comment. We have revised the argument (Lines 153-166).

Line 125 and following. Please make clear that it is inevitable that subtracting a multiple of Ca (even a much smaller one than you use) from a flattish record will inevitably lead to the pattern observed and to a similar pattern at each site so this should not be used as evidence supporting your decision. Thus for example in line 129 “This suggests that terrestrial CaSO₄....during glacials” is not really a correct inference.

We have removed this argument.

Line 137 paragraph. This is very confusing and doesn't quite make sense until one reads the SI. I suggest a shorter paragraph in which you say that one could also correct for other salts (see SI) as a sensitivity study without greatly affecting the result, but that this would undermine the argument about the Ca/SO₄ ratio.

We have revised this argument (Lines 186-193).

However please also discuss the implications if a large part of the Ca:SO₄ correlation does arise from atmospheric reactions and how this would affect the correction you make.

We have carried out sensitivity tests by changing the Ca:SO₄ ratio (Lines 193-196).

SI, line 52. I do not follow how you make this conclusion that “primary is ...major” and secondary is minor. By all means there may be a reaction with nitrate but it is not clear why this rules out a reaction with sulfate.

A reaction with sulphate cannot be ruled out completely, but it is much slower compared to a reaction with nitrate (Lines 163-165, 191-193). Therefore, we assumed that terrestrial gypsum was a major source of CaSO₄. We have added “the reaction with SO₂/H₂SO₄ is very slow” to SI Line 63.

SI, section on MSA. I am deeply unconvinced by this calculation. Your calculation multiplying MSA by 1/A seems entirely unjustified ? there is no mechanistic reason why this should be the case, nor for the choice of del18O threshold (which mechanistically should be a threshold in dust concentration). Of course with suitable tuning of A one can reproduce higher values in interglacials but this is entirely fabricated and seems to have no validity. I urge scrapping this section.

We have removed this section.

SI, lines 117-121. Please see my comments above on the relative roles of AZ and SAZ.

We have changed the word “dominant” to “major” (SI Line 94), as written above. We also added an explanation about the export production in SAZ (Lines Line 78-79).

Line 121. Surely the APF is the northern edge of the AZ not southern?

Thank you for the comment. It was an error. We corrected it.

Reviewer #3 (Remarks to the Author):

Review of Goto-Azuma et al, ‘Reduced marine phytoplankton sulphur emissions in the Southern Ocean during the past seven glacial periods’, submission to Nature Communications, July 2018.

This paper presents new ionic composition data of the Dome Fuji ice core, along with published data from the EPICA ice cores, to suggest there were glacial interglacial changes in ‘residual non-seasalt-sulfate’ which is interpreted to be related to biogenic sulfur emissions from the Southern Ocean. There is higher residual non-seasalt sulfate during interglacial periods than glacial periods and this is related to support other observations of higher marine productivity in the Antarctic zone during interglacials.

I think this manuscript is very interesting and understanding the connection between the marine sulfur cycle and global climate change is very important to thinking in the field. I think there needs to be more justification of exactly how the authors chose to calculate the residual non-seasalt sulfate before publication could be considered, as this differs from the previous studies that they cite of the EDC core.

In particular, the Kaufman et al. (2010, QSR) study cited evaluates how much of the sulfate could be contributed by mineral dust. In the Kaufman study, they assumed a maximum sulfate/calcium ratio of dust of 0.5 (mass ratio) based on measurements of Saharan dust (agreeably potentially not the most relevant material) and calculated a maximum of 16% mineral sulfate. The present paper assumed sulfate/calcium of dust = 2.4, almost 5 times higher, and this results in what looks like 75% mineral sulfate in some cases (hard to tell without having the tabulated data). In other words, the present paper is assuming all of the Ca in mineral dust is CaSO₄. The authors deduce that any CaCO₃ originally in the dust would have been converted to Ca(NO₃)₂ rather than sulfate. Fair enough.

But what if the majority of the Ca is as silicate minerals (clays, plagioclase, feldspar, etc) which contain Ca but no SO₄? The Sakurai et al. study they cite as determining all the Ca²⁺ is as gypsum also seems to measure a substantial fraction of feldspar (as 'other'), although that particular method may not be the best for determining calcium silicates. I think more justification needs to be given as to the mineralogical composition of the Patagonian dust and it should be clearly stated why the SO₄/Ca ratio they use is different from that which was assumed previously.

1. As Reviewer # 3 points out, Saharan dust is not a relevant material for our study. We have demonstrated significant source regions where gypsum is a major mineral (Lines 153-163). In some of the regions, the SO₄/Ca ratio is close to the stoichiometric ratio 2.4. We therefore believe that the SO₄/Ca ratio could be much higher than 0.5, which was suggested by Kaufman et al. as an uppermost ratio.

2. Iizuka et al. (2009) shows that a large fraction of the particles contained in the Dome Fuji LGM samples consisted of silicate minerals containing Ca, but that they did not dissolve in water. The majority of Ca²⁺ measured in this study using ion chromatography should therefore have originated from CaSO₄.

3. Although Sakurai et al detected feldspar, the feldspar they detected was Albite (NaAlSi₃O₈) (Sakurai's Ph.D. thesis).

One other study I came across while doing this review may be of use. Alexander et al. (2003, JGR doi:10.1029/2003JD003513) measured sulfur isotopes in Vostok and Dome C sulfate. This study found that the isotope composition of the sulfate potentially implied a large proportion of mineral sulfate (~50%). However, because an estimate of mineral sulfate using Ca, Na, Mg and NO₃, citing LeGrand et al 1988 Atmos Environ) suggested <10% mineral sulfate, these authors concluded that the isotope anomaly must therefore be reflecting a fractionation process rather than simple end-member mixing. Perhaps this study would need to be reinterpreted as well if the mineral sulfate really is as high as the present authors suggest.

We appreciate this comment very much. When Alexander et al. (2003) published the paper, they had to assume isotopic fractionation to explain 4-6‰ lower δ³⁴S for the last glacial with respect to the Holocene, because they considered terrestrial input was low. However, a recent study on surface snow samples from a latitudinal transect between a coastal station (Syowa) and interior site (Dome Fuji) suggests that isotopic fractionation during long-range transport is insignificant in East Antarctica and thus δ³⁴S in the ice cores from the East Antarctic interior can be used to infer source contributions (Uemura et al.2016). We have explained this in Introduction (Lines 82-93) and Results (Lines 203-204).

As Reviewer #3 points out, the continental source represents ~50% of the nss

sulphate if there is no fractionation. This ratio agrees well with that estimated from our residual nss sulphate. We have explained this in the text (Lines 199-213) and added Fig. S6.

We could not figure out how Alexander et al. estimated the contribution of mineral sulphate. We have read Legrand et al (1988) paper, but it was not clear.

If this major issue can be settled, the subsequent discussion of the possible causes and/or consequences of varying marine sulfur emissions are very valuable targets for future global climate research.

We hope that the major issue has been settled in the current version of the manuscript. If not, we would appreciate further comments from the reviewers.

Reviewers' comments:

Reviewer #1 (Remarks to the Author):

This paper has been revised very considerably since the previous version. The authors have acknowledged and incorporated a large number of the suggestions of both reviewers. The obvious difficulties with their hypothesis are at least now discussed. The paper makes a much better case for a large terrestrial origin for sulfate in ice, and I appreciate the incorporation of sulfur isotope evidence as proposed by both reviewers. I feel sure that the paper can be published in a form not too far from this one, but it will need one further round of revision, not least because there is one section (around line 199 and Fig S6) now that this reviewer could not follow. I also do feel that one issue with the authors' hypothesis remains and should be acknowledged more strongly than it is. I will explain the areas where I still have problems in order of text using line numbers from the final (non-tracked) pdf.

Abstract, line 40 "We propose a new proxy and show reduced biogenic". I think this is an awkward phrasing; it is really a new way of calculating rather than a new proxy. Can I suggest "By correcting for this, we make a revised calculation of biogenic sulfate, and find that the its flux is reduced in....".

Line 54 "are also important indicators of primary productivity in the Southern Ocean (SO), because they are closely related to atmospheric CO₂". I don't understand the use of the word "because" in this sentence. I think you might mean "are also indicators of primary productivity in the Southern Ocean (SO), which is important because it is closely related to atmospheric CO₂".

Line 161. Here you quote high ratios of sulfate to Ca in some soils from South America. However when I look at these papers, I see that these soils also have high ratios of Mg/Ca and Na/Ca, often >>1. This is clearly not the case for nssMg and nssNa in the Antarctic ice samples, so soils such as these cannot be the source of what needs to be close to a pure gypsum signal.

Lines 164-168. I think the evidence presented that reactions of CaCO₃ with sulfate is ineffective is not entirely relevant. Most of the studies referred to seem to look at aerosol rather close to the source; in contrast we are looking in Antarctica at material that has had days to weeks to react. At least some of the papers indeed explain why reaction of SO₂ with CaCO₃ dust might be slow (no acidity), but over the Southern Ocean won't most of the SO₂ be converted already to sulfuric acid? I am not saying you are wrong but I am not sure the papers you cite establish the case for the conditions in the Southern Ocean, and still have the feeling that the different seasonality of dust and sulfate might also be in play. I don't think a major change is needed but unless you found a statement specifically saying something about the reaction with sulfuric acid then I think a more nuanced statement would be "The reaction of CaCO₃ with SO₂ is slow", and then "If this can also be

extended to the different conditions over the Southern Ocean, then terrestrial gypsum could therefore be needed to explain the relationship between Ca and sulfate in Antarctic ice, and be a major CaSO_4 source”.

Line 196 – I appreciate you doing sensitivity studies but am not sure what the conclusion is. Are you saying that your conclusion remains if $R > 1.2$, but fails if $R < 1.2$? Please clarify this. But in any case 1.2 is still a lot higher than even the highest value previous authors found plausible (0.5). I think that you still need to balance better the difficulties with each of the theories (i) and (ii) around line 144. (i) still has a major problem in requiring the main source of dust to Antarctica to be something rather close to gypsum, and you really haven’t established that such a source exists. As pointed out above the South American studies you cite do not have ratios of Mg and Na to Ca that can be supported by Antarctic data. On the other hand you correctly point out that (ii) also has a problem because it’s a bit hard to see why the minimum of the sulfate/Ca ratio would follow the stoichiometric line as it does. I think this calls for you still to be more cautious in saying that your theory is right rather than an equally plausible (or implausible) idea; perhaps you could cover this partly by saying that “to validate our idea it will be important to establish what source areas could provide such a gypsum-rich source of dust”.

Para starting line 199, and Fig S6. I really had a hard time understanding this paragraph, and it needs to be rewritten. I still don’t get it or the figure even though I have re-read it several times. It seems all to be the wrong way round. In the interglacials, sulfate flux is higher and Ca flux is very low, so hardly any of the sulfate should be terrestrial, and yet this is where you say it is all terrestrial (ratio of terrestrial/total is 1). Similarly, if the sulfate to Ca ratio is 2.4, then the amount of sulfate that is terrestrial should be higher than if it is 1.3, but you show it as lower. I think I may be misunderstanding what you did but in that case please explain this further.

Line 232: “the summer sea ice extent around Antarctica at the LGM was only slightly greater than the present day”. This is not what Gersonde (ref 40) shows. Rather it shows a considerably greater sea ice extent in the Atlantic sector (perhaps enough to explain the change in concentration?), an increase but not as great in the Indian sector, and no data anywhere else. I am not convinced you can rule out this cause of the change in your revised biogenic sulfate signal.

Line 234. Surely the interglacial values are always higher than the glacial, so do you mean here the “interglacial:glacial” ratios are 1.3 (and same in subsequent lines). Please check and correct this.

Responses to Reviewer's comments

We thank Reviewer #1 for the constructive and helpful comments. Our responses to each comment are written in blue italicized characters.

Reviewer #1 (Remarks to the Author):

This paper has been revised very considerably since the previous version. The authors have acknowledged and incorporated a large number of the suggestions of both reviewers. The obvious difficulties with their hypothesis are at least now discussed. The paper makes a much better case for a large terrestrial origin for sulfate in ice, and I appreciate the incorporation of sulfur isotope evidence as proposed by both reviewers. I feel sure that the paper can be published in a form not too far from this one, but it will need one further round of revision, not least because there is one section (around line 199 and Fig S6) now that this reviewer could not follow. I also do feel that one issue with the authors' hypothesis remains and should be acknowledged more strongly than it is. I will explain the areas where I still have problems in order of text using line numbers from the final (non-tracked) pdf.

Thank you for pointing out this error. Indeed, we had mistakenly written "terrestrial / total nss SO_4^{2-} ratios" for R_2 in the section around line 199 and Fig. S6. We have corrected the text to read "marine biogenic / total nss SO_4^{2-} ratio".

Abstract, line 40 "We propose a new proxy and show reduced biogenic". I think this is an awkward phrasing; it is really a new way of calculating rather than a new proxy. Can I suggest "By correcting for this, we make a revised calculation of biogenic sulfate, and find that the its flux is reduced in....".

We appreciate the reviewer's recommendation and have corrected the phrasing as suggested.

Line 54 "are also important indicators of primary productivity in the Southern Ocean (SO), because they are closely related to atmospheric CO_2 ". I don't understand the use of the word "because" in this sentence. I think you might mean "are also indicators of primary productivity in the Southern Ocean (SO), which is important because it is closely related to atmospheric CO_2 ".

Thank you for pointing out the need for clarity here. We have corrected the phrasing as suggested.

Line 161. Here you quote high ratios of sulfate to Ca in some soils from South America. However when I look at these papers, I see that these soils also have

high ratios of Mg/Ca and Na/Ca, often $\gg 1$. This is clearly not the case for nssMg and nssNa in the Antarctic ice samples, so soils such as these cannot be the source of what needs to be close to a pure gypsum signal.

Thank you for making this important point. We have modified two paragraphs in the revised manuscript (one starting from Line 158 and the second starting from Line 169). We believe that the previously assumed small terrestrial $\text{SO}_4^{2-}/\text{Ca}^{2+}$ ratios cannot be taken for granted. We point out that the ratio could be much higher in the paragraph starting from Line 158. We are willing to further revise or remove this paragraph if the changes remain unclear.

Lines 164-168. I think the evidence presented that reactions of CaCO_3 with sulfate is ineffective is not entirely relevant. Most of the studies referred to seem to look at aerosol rather close to the source; in contrast we are looking in Antarctica at material that has had days to weeks to react. At least some of the papers indeed explain why reaction of SO_2 with CaCO_3 dust might be slow (no acidity), but over the Southern Ocean won't most of the SO_2 be converted already to sulfuric acid? I am not saying you are wrong but I am not sure the papers you cite establish the case for the conditions in the Southern Ocean, and still have the feeling that the different seasonality of dust and sulfate might also be in play. I don't think a major change is needed but unless you found a statement specifically saying something about the reaction with sulfuric acid then I think a more nuanced statement would be "The reaction of CaCO_3 with SO_2 is slow", and then "If this can also be extended to the different conditions over the Southern Ocean, then terrestrial gypsum could therefore be needed to explain the relationship between Ca and sulfate in Antarctic ice, and be a major CaSO_4 source".

This is correct. Most referenced studies do look at aerosols rather close to the source, and we agree that conditions over the Southern Ocean could be different. However, we do not think there is sufficient information to discuss where SO_2 is converted to sulfuric acid in the Southern Ocean. Carrico et al. (reference No. 34) further report that in volcanic-dominated air masses near Japan, dust from China was only partially neutralized in the presence of H_2SO_4 and SO_2 . We agree with the point made by de Angelis et al. (reference No. 20) that this suggests the reaction between CaCO_3 and H_2SO_4 (as well as with SO_2) is slow; however, we are not certain that this applies to conditions over the Southern Ocean. Therefore, to address the reviewer's helpful suggestion, we have added "If this can also be extended to the different conditions over the Southern Ocean, then terrestrial gypsum would be needed to explain the relationship between nss Ca^{2+} and nss SO_4^{2-} in Antarctic ice core and may be a major CaSO_4 source." Please see Lines 181-183 in the revised manuscript.

We tried but were unable to find evidence of different seasonality of dust and sulphate. Wegener et al. (JGR, 2015) review previous studies on dust seasonality and conclude that these studies do not provide a consistent picture

for the entire Antarctic continent. Although dust concentration maxima occur in the winter at EDML, different sites in Antarctica show different seasonalities. At Law Dome, elevated concentrations were found in the spring and autumn with a minimum in the winter (Burn-Nunes et al., 2011). The present day emission strength of the South American source is highly variable but shows a maximum in the summer (Johnson et al., 2010). We therefore do not argue that different seasonalities of dust and sulphate would make the reaction between CaCO_3 and sulphate difficult.

Line 196 ? I appreciate you doing sensitivity studies but am not sure what the conclusion is. Are you saying that your conclusion remains if $R_1 > 1.2$, but fails if $R_1 < 1.2$? Please clarify this. But in any case 1.2 is still a lot higher than even the highest value previous authors found plausible (0.5). I think that you still need to balance better the difficulties with each of the theories (i) and (ii) around line 144. (i) still has a major problem in requiring the main source of dust to Antarctica to be something rather close to gypsum, and you really haven't established that such a source exists. As pointed out above the South American studies you cite do not have ratios of Mg and Na to Ca that can be supported by Antarctic data. On the other hand you correctly point out that (ii) also has a problem because it's a bit hard to see why the minimum of the sulfate/Ca ratio would follow the stoichiometric line as it does. I think this calls for you still to be more cautious in saying that your theory is right rather than an equally plausible (or implausible) idea; perhaps you could cover this partly by saying that "to validate our idea it will be important to establish what source areas could provide such a gypsum-rich source of dust".

Thank you for pointing out the need for clarity here. The statement is correct that our conclusion remains if $R_1 > 1.2$ but fails if $R_1 < 1.2$. Indeed, 1.2 is considerably higher than the highest previously reported value (0.5). However, we do not agree that 0.5 is the highest plausible value. We do not think that the Saharan dust ratio can constrain the ratio in South America because Bouza et al. (1993, 2007) show that some of the Patagonian soil samples have substantially higher ratios, although these soils may not be the source for Antarctic ice core sites, as commented by the reviewer. Because we believe that the previously assumed small terrestrial $\text{SO}_4^{2-}/\text{Ca}^{2+}$ ratios cannot be taken for granted, we have pointed out (in the paragraph starting from Line 158) that the ratio could be much higher than previously assumed. However, we are willing to further revise or remove this paragraph if deemed necessary.

We appreciate the reviewer's suggestion and have added the suggested text in Lines 184–185.

Para starting line 199, and Fig S6. I really had a hard time understanding this paragraph, and it needs to be rewritten. I still don't get it or the figure even though I have re-read it several times. It seems all to be the wrong way round. In

the interglacials, sulfate flux is higher and Ca flux is very low, so hardly any of the sulfate should be terrestrial, and yet this is where you say it is all terrestrial (ratio of terrestrial/total is 1). Similarly, if the sulfate to Ca ratio is 2.4, then the amount of sulfate that is terrestrial should be higher than if it is 1.3, but you show it as lower. I think I may be misunderstanding what you did but in that case please explain this further.

We are grateful to the reviewer for pointing out this unintended and important error around line 199 and Fig S6. We had erroneously written “terrestrial / total nss SO₄²⁻ ratio” but have corrected this error in the text and figure caption to read “marine biogenic / total nssSO₄²⁻ ratio”.

Line 232: “the summer sea ice extent around Antarctica at the LGM was only slightly greater than the present day”. This is not what Gersonde (ref 40) shows. Rather it shows a considerably greater sea ice extent in the Atlantic sector (perhaps enough to explain the change in concentration?), an increase but not as great in the Indian sector, and no data anywhere else. I am not convinced you can rule out this cause of the change in your revised biogenic sulfate signal.

Thank you for pointing out the need for clarity here. We have made the following changes to more carefully refer to the Gersonde et al. study. “In the Indian Ocean sector, the summer sea ice extent at the LGM was only slightly greater than the present day, whereas in the Atlantic Ocean sector, the sporadic occurrence of summer sea ice considerably farther north is indicated (Supplementary Discussion). Although data from other oceanic sectors are very sparse, Gersonde et al. speculates that the summer sea ice field around Antarctica changed from 4×10^6 km² (present day) to $5\text{--}6 \times 10^6$ km² (LGM).” We have also added a few additional sentences in the following paragraph.

Line 234. Surely the interglacial values are always higher than the glacial, so do you mean here the “interglacial:glacial” ratios are 1.3 (and same in subsequent lines). Please check and correct this.

Thank you for pointing out this unintended error. We meant that glacial/interglacial ratios are 1/3–1/4. We have revised this sentence accordingly.

In addition to revisions made following the reviewer’s comments, we have also made the following minor revisions:

- *We have added a brief summary in the last paragraph of the “Introduction”, following the “Manuscript Checklist”;*
- *We have changed the subheadings of the “Results” section;*
- *We have added a “Discussion” section and relocated some sentences from*

the “Results” to the “Discussion” section;

- *We have changed the order of Supplementary Figs. 4 and 5;*
- *We have corrected the caption of Supplementary Fig. 7 because $ssNa^+$ was accidentally written as $nssNa^+$;*
- *We have added one sentence to the captions of Supplementary Figs. 4 and 7;*
- *We have added a “Spectral analysis” sub-section in the “Methods” section;*
- *We have made some minor changes to improve the English.*

REVIEWERS' COMMENTS:

Reviewer #1 (Remarks to the Author):

Thank you for again making significant changes that clarify your argument while also making clear where uncertainties may undermine your hypothesis. I am happy now that the paper is a valuable contribution with a genuinely new idea, well explained. It may or may not turn out to be correct but the discrepancies in the previous interpretation are significant enough that this deserves to be published and discussed. I recommend publication. There are a few places where some editing of the language may still be required but overall the manuscript is in good shape.